# Zebrafish *mylipb* attenuates antiviral innate immunity through two synergistic mechanisms targeting transcription factor irf3

Zhi Li[1,2], Jun Li[1,3], Ziyi Li[1,3], Yanan Song[1,3], Yanyi Wang[1,3], Chunling Wang[1], Le Yuan[1,3], Wuhan Xiao[1,3,4,5]*, Jing Wang[1,3,4]*

1 State Key Laboratory of Freshwater Ecology and Biotechnology, Institute of Hydrobiology, Chinese Academy of Sciences, Wuhan, China, 2 State Key Laboratory of Biocatalysis and Enzyme Engineering, School of Life Sciences, Hubei University, Wuhan, China, 3 University of Chinese Academy of Sciences, Beijing, China, 4 Hubei Hongshan Laboratory, Wuhan, China, 5 The Innovation of Seed Design, Chinese Academy of Sciences, Wuhan, China

* w-xiao@ihb.ac.cn (WX); wangjing@ihb.ac.cn (JW)

**Data Availability Statement:** The authors confirm that all data underlying the findings are fully available without restriction. All relevant data are

## Abstract

IFN regulatory factor 3 (IRF3) is the transcription factor crucial for the production of type I IFN in viral defence and inflammatory responses. The activity of IRF3 is strictly modulated by post-translational modifications (PTMs) to effectively protect the host from infection while avoiding excessive immunopathology. Here, we report that zebrafish myosin-regulated light chain interacting protein b (*mylipb*) inhibits virus-induced type I IFN production via two synergistic mechanisms: induction of autophagic degradation of irf3 and reduction of irf3 phosphorylation. In *vivo*, *mylipb*-null zebrafish exhibit reduced lethality and viral mRNA levels compared to controls. At the cellular level, overexpression of *mylipb* significantly reduces cellular antiviral capacity, and promotes viral proliferation. Mechanistically, mylipb associates with irf3 and targets Lys 352 to increase K6-linked polyubiquitination, dependent on its E3 ubiquitin ligase activity, leading to autophagic degradation of irf3. Meanwhile, mylipb acts as a decoy substrate for the phosphokinase tbk1 to attenuate irf3 phosphorylation and cellular antiviral responses independent of its enzymatic activity. These findings support a critical role for zebrafish *mylipb* in the limitation of antiviral innate immunity through two synergistic mechanisms targeting irf3.

## Author summary

As an indispensable transcription factor for antiviral innate immunity, IRF3 activity is tightly regulated by post-translational modifications to effectively protect the host from infection while avoiding excessive immunopathology. In this study, we report that zebrafish *mylipb* is induced by viral stimulation, which negatively regulates irf3-mediated antiviral innate immunity. In *mylipb*[-/-], the antiviral capacity and IFN response are significantly higher than that in *mylipb*[+/+]. Furthermore, mylipb negatively regulates antiviral innate immunity by two synergistic mechanisms: mylipb promotes K6-linked polyubiquitination of irf3 at lysine 352; mylipb acts as a decoy substrate for the phosphokinase

within the paper and its Supporting Information files.

**Funding:** This work was supported by grants from The Strategic Priority Research Program of the Chinese Academy of Sciences (XDA24010308 to WX); by the National Natural Science Foundation of China (31972786 to JW and 31830101 to WX); by the Innovative Research Group Project of the National Natural Science Foundation of China (31721005 to WX). The funders had no role in study design, data collection and analysis, decision to publish, or preparation of the manuscript.

**Competing interests:** The authors have declared that no competing interests exist.

tbk1 to attenuate irf3 phosphorylation. Our findings expand our knowledge of the negative regulatory mechanisms of IRF3 and reveal a novel function of *mylipb* in innate antiviral responses.

## Introduction

Innate immunity represents an evolutionary conserved mechanism that serves as the host's initial defence against invading pathogens of microbial pathogens [1–3]. The recognition of conserved pathogen-associated molecular patterns (PAMPs) by cellular pattern recognition receptors (PRRs) is essential for the initiation of the innate antiviral immune response [4–8]. Virus-derived nucleic acids are mainly recognized by the DNA sensor cyclic guanosine monophosphate-adenosine monophosphate synthase (cGAS) and the RNA sensors retinoic acid-inducible gene 1 (RIG-I)-like receptors (RLRs), including RIG-I and melanoma differentiation-associated gene 5 (MDA5), and laboratory of genetics and physiology 2 [LGP2, 9–14]. Activation of the sensors recruits the signaling adaptor proteins, including STING (stimulator of interferon [IFN] genes protein) and MAVS (mitochondrial antiviral signalling protein), which then activate the downstream signaling molecules TANK binding kinase 1 [TBK1, 15–19]. These cascades eventually converge on the activation of the transcription factor IFN regulatory factor 3 (IRF3), which serves as an indispensable transcription factor for type I IFN production and subsequent ISGs for host antiviral state [20–22]. Upon receiving upstream signals, IRF3 undergoes several processes to activate the type I IFN pathway, including phosphorylation, dimerisation, nuclear translocation, promoter binding and facilitating transcriptional activation of type I IFN genes [21–23].

Post-translational modifications (PTMs), such as phosphorylation, ubiquitination, acetylation, deamidation, and methylation have been shown to influence PRRs-dependent immune responses via their modifications of receptors, adaptors, kinase, and transcriptional factors [24–28]. In particular, ubiquitination plays an indispensable role in fine-tuning the RLR-mediated antiviral response [29–33]. Ubiquitin ligases serve as key regulators of this signaling by precisely regulating the dynamic ubiquitination process [29,34,35]. The ubiquitin molecule itself can be ubiquitinated at seven internal lysine residues (K6, K11, K27, K29, K33, K48 or K63) to produce the corresponding polyubiquitination modifications [34]. Among these, polymeric chains based on K48 and K63 have been extensively studied. For example, K48-linked polyubiquitination is the canonical signal that mediates protein degradation in a proteasome manner, whereas K63-linked polyubiquitination mainly contributes to signal transduction, DNA repair enzymatic activity and protein endocytosis [36]. Although K48- and K63-linked polyubiquitination are widely recognized to influence innate antiviral immune responses, other atypical polyubiquitination modes such as K6-, K11-, K27-, K29-, K33- and linear-linked polyubiquitination are being investigated for their distinct roles in innate immunity [37–40]. As a key transcriptional factor in IFN-I signaling, IRF3 activity has been shown to be regulated by ubiquitination and deubiquitination [38,41–46]. Meanwhile, IRF3 phosphorylation is required for its homodimerisation and nuclear translocation, and the mechanisms underlying phosphorylation-triggered activation of IRF3 have been the subject of many extensive studies [18,47]. Understanding the mechanisms that regulate IRF3 phosphorylation and ubiquitination has been an interesting topic in the field of innate immunity.

Myosin regulatory light-chain interacting protein (MYLIP, also known as inducible degrader of the low-density lipoprotein receptor) is currently the only really interesting new gene (RING) family ubiquitin ligase E3 with FERM (Band 4.1, ezrin-radixin-moesin) domain

[48,49]. The function of MYLIP has been reported to monitor cellular and organismal cholesterol homeostasis by promoting the ubiquitin-mediated degradation of lipid metabolism proteins such as low-density lipoprotein receptor (LDLR), very-low-density lipoprotein receptor (VLDLR), and apolipoprotein E receptor 2[ApoER2, 50–55]. In addition, MYLIP can modulate its own stability by forming a homodimer followed by autoubiquitination and degradation via the proteasome pathway [56,57]. However, the role of MYLIP in vertebrate virus-induced innate immunity signaling is still largely unknown.

In this study, we report that *mylipb*-null zebrafish are hyposensitive to SVCV infection and exhibit an enhanced induction of the IFN response. Our in *vitro* data suggest that zebrafish mylipb associates with irf3 and increases K6-linked polyubiquitination at Lys 352, leading to the induction of autophagic degradation of irf3. In addition, mylipb acts as a decoy substrate for the phosphokinase tbk1 to attenuate irf3 phosphorylation. Our results demonstrate that zebrafish *mylipb* is a negative regulator of type I IFN signaling and the cellular antiviral response, with two synergistic mechanisms working together to target irf3.

## Results

### Zebrafish *mylipb* is induced by poly I:C treatment and SVCV infection

In zebrafish, there are two orthologous genes of *MYLIP*, including *mylipa* and *mylipb*. To know the behaviour of zebrafish *mylip* in response to viral infection, ZFL cells were stimulated with poly I:C, a mimic of RNA virus [58], for 0–48 h. Using quantitative real-time PCR (qRT-PCR) assays, we found that similar to the key antiviral gene, *ifn1* [59], mRNA expression of *mylipb* was more significantly upregulated than *mylipa* (Fig 1A). Consistently, we challenged ZFL cells with SVCV for 0–48 h and found that *mylipb* expression was also increased more notably than *mylipa* (Fig 1B). *MYLIP* is transcriptionally induced by the liver X receptor (LXR), a transcription factor involved in the regulation of cholesterol metabolism [55]. We then challenged ZFL with T0901317, an LXR agonist, for 24 h and found that only *mylipa* expression was significantly upregulated at different doses (Fig 1C). However, *mylipb* expression was barely affected by T0901317. A similar trend was observed during the time gradient treatment with T0901317 (Fig 1D). Therefore, we focused our subsequent assays on *mylipb* function.

### *Mylipb* exhibits proviral effects both in *vivo* and in *vitro*

To understand the physiological role of *mylipb* in the antiviral response in *vivo* and in *vitro*, we first generated *mylipb*-deficient zebrafish using CRISPR-Cas9 and obtained a mutant line with a 7-nucleotide deletion in exon 3 of *mylipb* (S1A and S1B Fig). Crossing *mylipb*$^{+/-}$ resulted in offspring with *mylipb*$^{+/+}$, *mylipb*$^{+/-}$ and *mylipb*$^{-/-}$ genetic backgrounds in a Mendelian ratio (1:2:1), and no obvious defects in growth rate and reproductive ability were detected in *mylipb*$^{-/-}$ zebrafish under normal conditions. qRT-PCR assay confirmed that *mylipb* was efficiently disrupted (S1C Fig). We challenged *mylipb*$^{-/-}$ larvae (3 days post-fertilisation [dpf]) and *mylipb*$^{+/+}$ larvae (3 dpf) (having a wild-type allele of mylipb) (WT) with high titer SVCV and photographed the larvae after 18 h (Fig 2A). We found that *mylipb*-deficient zebrafish larvae were more resistant to SVCV-induced death (Fig 2A and 2B). We then injected SVCV intraperitoneally (i.p.) into *mylipb*$^{-/-}$ and *mylipb*$^{+/+}$ adult zebrafish (3 months post-fertilisation [mpf]) and used cell culture medium as a control, and subsequently observed their phenotypes. Compared to SVCV-injected WT zebrafish, SVCV-injected *mylipb*-deficient zebrafish showed less abdominal swelling and haemorrhagic symptoms (Fig 2C). In addition, we counted dead zebrafish at different time points after SVCV injection and plotted a survival curve. As shown in Fig 2D, after challenge with SVCV, *mylipb*$^{-/-}$ zebrafish showed a higher survival rate

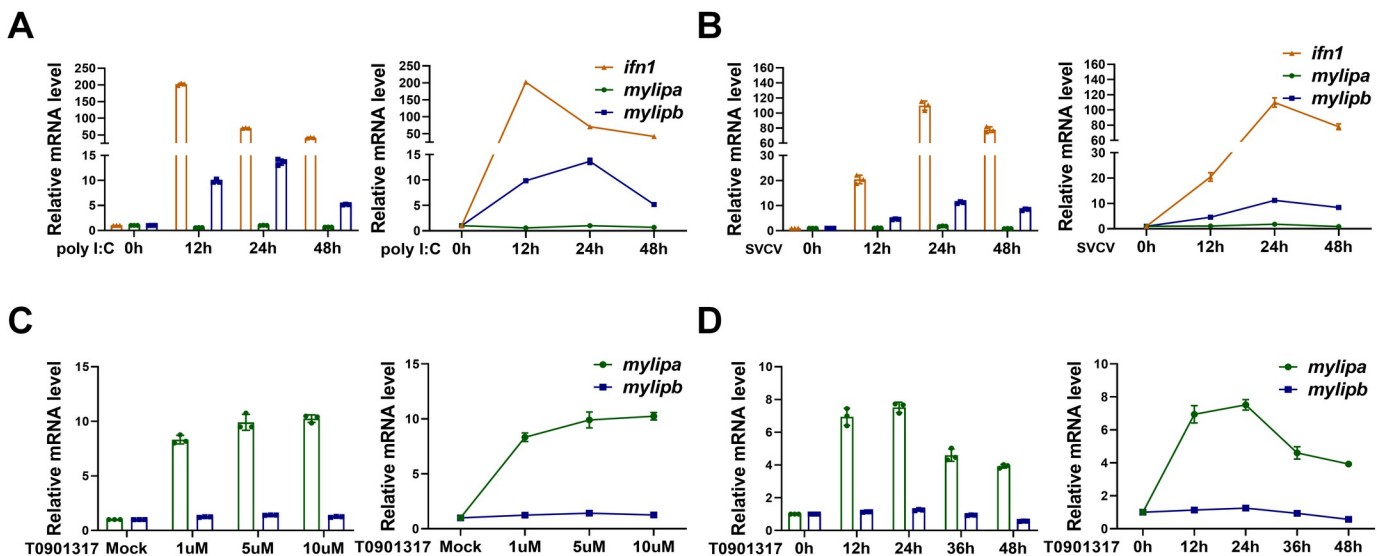

**Fig 1. Zebrafish *mylipb* is upregulated upon poly I:C treatment and SVCV infection.** (A) Zebrafish *mylipb* as well as *ifn1*, but not *mylipa* mRNA was induced by poly I:C in ZFL cells. ZFL cells were seeded in 60-mm plates overnight and treatment with poly I:C (1 μg/mL) for indicate time. (B) Zebrafish *mylipb* as well as *ifn1*, but not *mylipa* mRNA was induced by SVCV (virus strain OMG067) infection (MOI of 1) in ZFL cells. ZFL cells were seeded in 60-mm plates overnight and infected with SVCV for different time. (C) Zebrafish *mylipa* but not *mylipb* mRNA was induced by LXR agonist T0901317 in a dose-dependent manne in ZFL cells. ZFL cells were seeded in 60-mm plates overnight and treatment with T0901317 for 24 h. (D) Zebrafish *mylipa* but not *mylipb* mRNA was induced by LXR agonist T0901317 (1 μM) in a indicate time in ZFL cells. ZFL cells were seeded in 60-mm plates overnight and treatment with T0901317 for indicate time. Total RNA was extracted for examining the expression of target genes by quantitative real-time PCR (qRT-PCR). All data are presented as mean values based on three repeated experiments, and error bars indicate the ± SD. *,P < 0.05, **, P<0.01; ***, P < 0.001; ****, P< 0.0001.

compared to WT zebrafish (Fig 2D). Consistently, *mylipb*$^{-/-}$ larvae exhibited lower mRNA expression of N-, P- and G-protein genes of SVCV genes than WT larvae (Fig 2E).

To evaluate the alteration of IFN response by *mylipb* deficiency, we compared the mRNA expression of three antiviral genes, including *ifn1*, *mxc* and *lta* [60], between *mylipb*$^{-/-}$ and WT larvae following SVCV infection. qRT-PCR assay showed that the transcription of these antiviral genes induced by SVCV was significantly increased in *mylipb*$^{-/-}$ larvae compared to WT larvae (Fig 2F–2H). Inflammatory genes also contribute to the viral pathogenesis, we also compared the mRNA expression of inflammation factor, including *il-1β*, *il8*, *tnfα*, between *mylipb*$^{-/-}$ and WT larvae following SVCV infection. qRT-PCR assay showed that the transcription of these genes induced by SVCV was significantly decreased in *mylipb*$^{-/-}$ larvae compared to WT larvae (S2A Fig). In addition, upon SVCV infection at the cellular level in *vitro*, mylipb significantly inhibited the antiviral capacity and enhanced the cytopathogenic effect (CPE) (Fig 2I). Next, dramatically increased viral titers and upregulated viral gene transcriptions (*N*, *P* and *G* genes) were found in *mylipb*-overexpressing EPC cells compared to controls (Fig 2J–2K). Taken together, these findings indicate that mylipb is a host negative factor against viral infections in *vivo* and in *vitro*.

## *Mylipb* inhibits poly I:C and SVCV-induced IFN activation

Since IFN activation plays a key role in the host's antiviral capacity, it is necessary to further investigate the effect of *mylipb* on IFN response. Using promoter assays, we found that overexpression of *mylipb* suppressed the ISRE reporter (a commonly used reporter for monitoring viral infection)[61] induced by either poly I:C transfection or SVCV infection in EPC cells (Fig 3A and 3B). Consistent with the ISRE reporter results, mylipb significantly inhibited Dr-IFNφ1-luc (the zebrafish ifnφ1 promoter) activity induced by poly I:C or SVCV infection

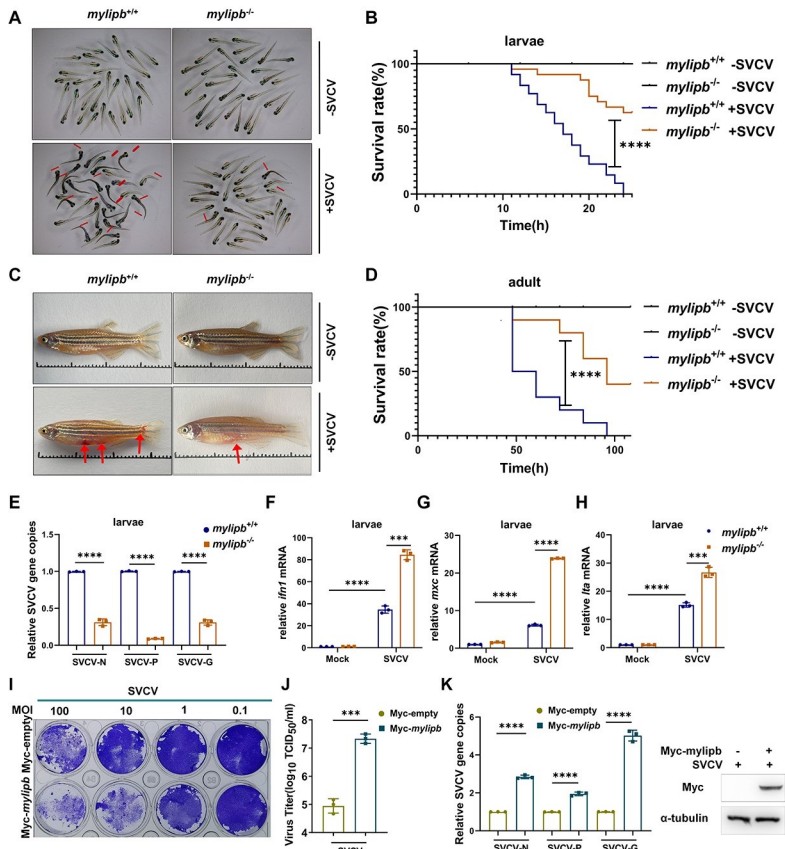

**Fig 2. Zebrafish *mylipb* exhibits proviral effects both in vivo and in vitro.** (A) Representative images of *mylipb*-null zebrafish larvae (*mylipb⁻/⁻*) (3 dpf) and WT siblings (*mylipb⁺/⁺*) (3 dpf) infected with or without SVCV for 18 h. Zebrafish larvae (3 dpf) were pooled in a disposable 60-mm cell culture dish filled with 5mL of egg water plus 2 mL SVCV ($\sim 2.5 \times 10^7$ TCID$_{50}$/ml). Dead larvae were identified by lack of movement, absence of blood circulation, and bodily degeneration (indicated by red arrows). (B) *mylipb*-null zebrafish (*mylipb⁻/⁻*) (n = 48, 3 dpf) were more resistance to SVCV infection than the WT (*mylipb⁺/⁺*) (n = 48, 3 dpf) based on the survival ratio. SVCV viruses ($\sim 2.50 \times 10^7$ TCID$_{50}$/mL) were added to *mylipb*-null and WT larvae, and the numbers of dead larvae were counted every hour from 0 to 24 h. (C) *mylipb*-null zebrafish adults are more resistance to SVCV infection than WT zebrafish. *mylipb*-null zebrafish (*mylipb⁻/⁻*)(3 months postfertilization [3 mpf]) displayed less symptoms of abdominal hemorrhage after SVCV infection than WT zebrafish (*mylipb⁺/⁺*)(3 mpf). The zebrafish were i.p. injected with either 10 μL cell culture medium or 10 μL SVCV($\sim 2.50 \times 10^7$ TCID$_{50}$/mL). The red arrows indicate hemorrhage regions. (D) *mylipb*-null zebrafish (*mylipb⁻/⁻*) (3 mpf) (n = 10) were more resistance to SVCV infection than the WT (*mylipb⁺/⁺*) (3 mpf) (n = 10) based on the survival ratio. (E) The virus replication number was lower in *mylipb*-null zebrafish larvae than in the WT zebrafish after infection with SVCV. *mylipb*-null larvae (*mylipb⁻/⁻*) and the WT larvae (*mylipb⁺/⁺*) are offspring of siblings. 30 larvae (3 dpf) were pooled in a disposable 60-mm cell culture dish filled with 5 mL of egg water and 0.5 mL of SVCV ($\sim 2.5 \times 10^7$ TCID$_{50}$/ml).. After challenge for 24 h, the mRNA expression of N protein, P protein, and G protein of SVCV was detected by qRT-PCR. (F-H) The induction of key antiviral genes, including *mxc* (F), *ifn1* (G), *lta* (H) upon SVCV infection was higher in *mylipb*-null larvae (*mylipb⁻/⁻*) compared with the WT larvae (*mylipb⁺/⁺*). (I) Overexpression of *mylipb* reduced cell survival after SVCV infection in EPC cells. EPC cells were transfected with Myc-*mylipb* or empty vector. At 24 h post-transfection, the cells were infected with SVCV at the dose indicated for 48 h. Then, the cells were fixed with 4% paraformaldehyde and stained with 1% crystal violet. (J) Overexpression of *mylipb* increased virus titer in SVCV-infected EPC cells. The culture supernatant was collected from EPC cells infected with SVCV (MOI of 1), and the viral titer was measured by plaque assay. The results are representative of three independent experiments. (K) Overexpression of *mylipb* increased copy number of SVCV-related genes in SVCV-infected EPC cells. EPC cells were transfected with Myc-*mylipb* or empty vector for 24 h and infected with SVCV (MOI of 1). After 24 h, total RNAs were extracted for examining the mRNA levels of the *N*, *P*, and *G* gene of SVCV by qRT-PCR analysis. Western blotting tests to detect the overexpression of *mylipb*. All data are presented as mean values based on three repeated experiments, and error bars indicate the ± SD. *,$P < 0.05$, **, $P<0.01$; ***, $P < 0.001$; ****, $P< 0.0001$.

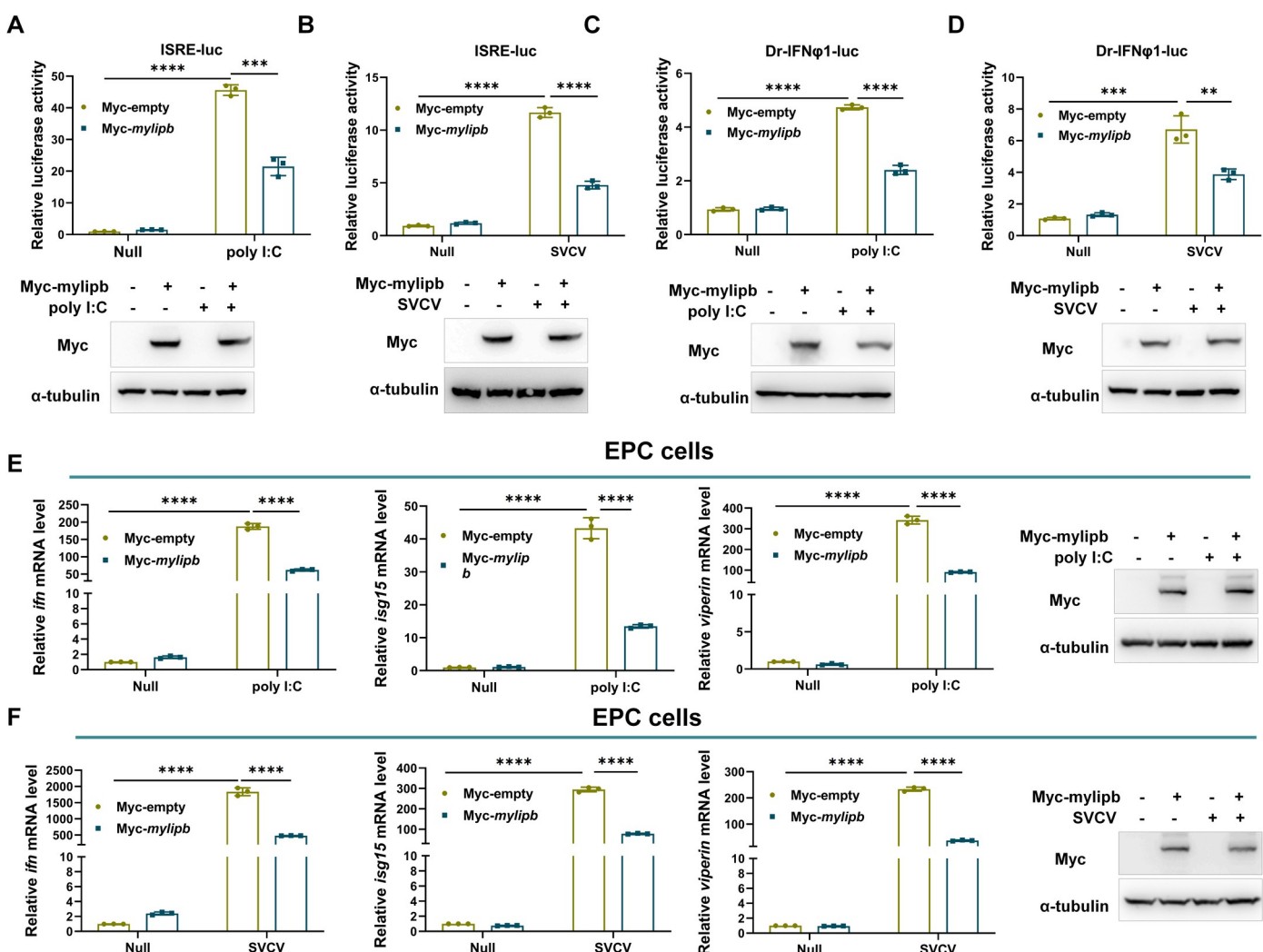

**Fig 3. Zebrafish *mylipb* negatively regulates IFN-I signaling.** (A-B) Overexpression of *mylipb* suppressed the activity of ISRE reporter (A), zebrafish IFNφ1 reporter (Dr-IFNφ1-Luc.) (B), induced by poly I:C transfected EPC cells. We transfected EPC cells with the indicated reporters (0.1 μg/well) along with Myc-empty vector or Myc-*mylipb* vector (0.2 μg/well). After 24 h, we transfected the cells with poly I:C (1 μg/mL) for 24 h and then conducted luciferase reporter activity and western blot assays. (C-D) Overexpression of *mylipb* suppressed the activity of ISRE reporter (C), zebrafish IFNφ1 reporter (Dr-IFNφ1-Luc.) (D), induced by SVCV infection in EPC cells. We transfected EPC cells with the indicated reporters (0.1 μg/well) along with Myc-empty vector or Myc-*mylipb* vector (0.2 μg/well). After 24 hours, we transfected the cells with SVCV (MOI of 1) for 24 hours and then conducted luciferase reporter activity and western blot assays. (E) Overexpression of *mylipb* suppressed expression of *ifn*, *isg15*, and *viperin* induced by poly I:C in EPC cells. EPC cells were transfected with Myc-*mylipb* or empty vector. After 24 hours, we transfected the cells with poly I:C (1 μg/mL) for 24 hours, total RNAs were extracted for examining the mRNA levels of *ifn*, *isg15*, and *viperin* by qRT-PCR analysis. Western blotting tests to detect the overexpression of *mylipb*. (F) Overexpression of *mylipb* suppressed expression of *ifn*, *isg15*, and *viperin* induced by SVCV infection in EPC cells. EPC cells were transfected with Myc-*mylipb* or empty vector for 24 h and infected with SVCV (MOI of 1). After 24 h, total RNAs were extracted for examining the mRNA levels of *ifn*, *isg15*, and *viperin* by qRT-PCR analysis. Western blotting tests to detect the overexpression of *mylipb*. All data are presented as mean values based on three repeated experiments, and error bars indicate the ± SD. *,$P < 0.05$, **, $P<0.01$; ***, $P < 0.001$; ****, $P< 0.0001$.

(Fig 3C and 3D). Subsequently, the effect of *mylipb* on IFN system transcription was also monitored. Consistent with the promoter assays, overexpression of *mylipb* strongly reduced the expression of *ifn* and two typical IFN-stimulated genes (*viperin* and *isg15*, 60) in EPC cells upon poly I:C or SVCV stimulation (Fig 3E and 3F). In addition, overexpression of *mylipb* up-regulated the expression of inflammatory genes *il1-β*, *il8*, *tnfα* in EPC cells upon SVCV stimulation (S2B Fig). These results demonstrate that mylipb acts as a suppressor of IFN activation.

## Mylipb associates with and induces autophagic degradation of irf3

There is increasing evidence that the fish RLR signalling pathway plays a critical role in the activation of IFN production [3,13,62]. In order to determine the mechanisms of zebrafish *mylipb* in the negative regulation of the antiviral response, we examined the activity of the ISRE reporter and EPC-IFN-luc stimulated by overexpressing of key molecules of the RLR pathway, including *rig-i*, *mda5*, *mavs*, *tbk1* and *irf3* together with Myc-empty vector or Myc-tagged *mylipb*. Overexpression of the RLR signaling cascade factors significantly activated the activity of the ISRE reporter and EPC-IFN-luc (Fig 4A and 4B). However, co-expression of *mylipb* suppressed the activity induced by these factors, from upstream to downstream factors in the RLR signaling pathway (Fig 4A and 4B). These data suggest that *mylipb* may negatively regulate antiviral immunity by suppressing the transactivity of irf3.

To further investigate whether irf3 activity was directly affected by mylipb, we examined their protein-protein interactions. As shown in Fig 4C, immunofluorescence staining confirmed the predominant colocalization of mylipb and irf3 protein in the cytoplasm (Fig 4C). In addition, co-immunoprecipitation assays further demonstrated that ectopically expressed mylipb pulled down irf3 in HEK293T cells (Figs 4D and S3A). Previous studies have shown that MYLIP exerts its physiological function by inhibiting the stabilization of target proteins [54]. Therefore, we further investigated whether the inhibition of the RLR pathway by mylipb was achieved by affecting the stability of the irf3 protein. As expected, overexpression of *mylipb* resulted in a dose-dependent reduction of overexpressed irf3 protein (Fig 4E). Upon SVCV infection, irf3 protein level and phosphorylation of irf3 were detected, which appeared to be higher in *mylipb*-null zebrafish (*mylipb*[−/−]) compared with WT zebrafish (*mylipb*[+/+]) (Fig 4F).

The degradation mechanism of irf3 in this process was then investigated. Protein degradation is mediated by three classical mechanisms, namely the proteasome, autophagosome and lysosomal proteolysis pathways, which are disrupted by MG132, 3-methyladenine (3-MA) or Baf-A1 and $NH_4Cl$, respectively. As shown in Fig 4G, MG132, a proteasomal inhibitor, had no effect on mylipb-induced irf3 protein degradation. However, 3-MA and Baf-A1, two autophagy inhibitors, clearly restored mylipb-induced irf3 protein degradation (Fig 4G and 4H), suggesting that mylipb might mediate autophagic degradation of irf3 protein. As expected, the addition of $NH_4Cl$ also restored mylipb-induced irf3 protein degradation, since the late stage of autophagy is involved in the lysosomal pathway (Fig 4I). Furthermore, overexpression of *mylipb* dose-dependently decreased the expression irf3 and was found to increase the levels of LC3-II (Fig 4J). Furthermore, during autophagy, conjugated LC3 coats the autophagosome membrane, so the observation of LC3 puncta formation is taken as an autophagy indicator. Confocal microscopy analysis revealed that LC3 puncta were significantly accumulated in cells overexpressing mylipb and irf3 (Fig 4K). To further strengthen this conclusion, HEK293T knockout cells for the essential autophagy genes ATG5 and BECN1 were used, and we found that the irf3 degradation induced by mylipb was detectable in wild-type (WT) cells but not in either ATG5 KO or BECN1 KO cells (S3B and S3C Fig).

Collectively, these data suggest that zebrafish mylipb induces autophagy-dependent degradation of irf3 to suppress RLR signaling activation.

## Mylipb degradates irf3 dependent of its ubiquitin ligase activity

To further elucidate the mechanisms of mylipb's effect on irf3 protein stability, we first performed domain mapping assays. The results showed that the zine finger (ZF) domain of mylipb is critical for its association with irf3 (Fig 5A and 5B). The cysteine residue at position 387 (C387) of the ZF domain is the enzymatic activity centre of MYLIP and necessary for maintaining the E3 ubiquitin ligase activity [55]. Amino acid sequence alignments revealed

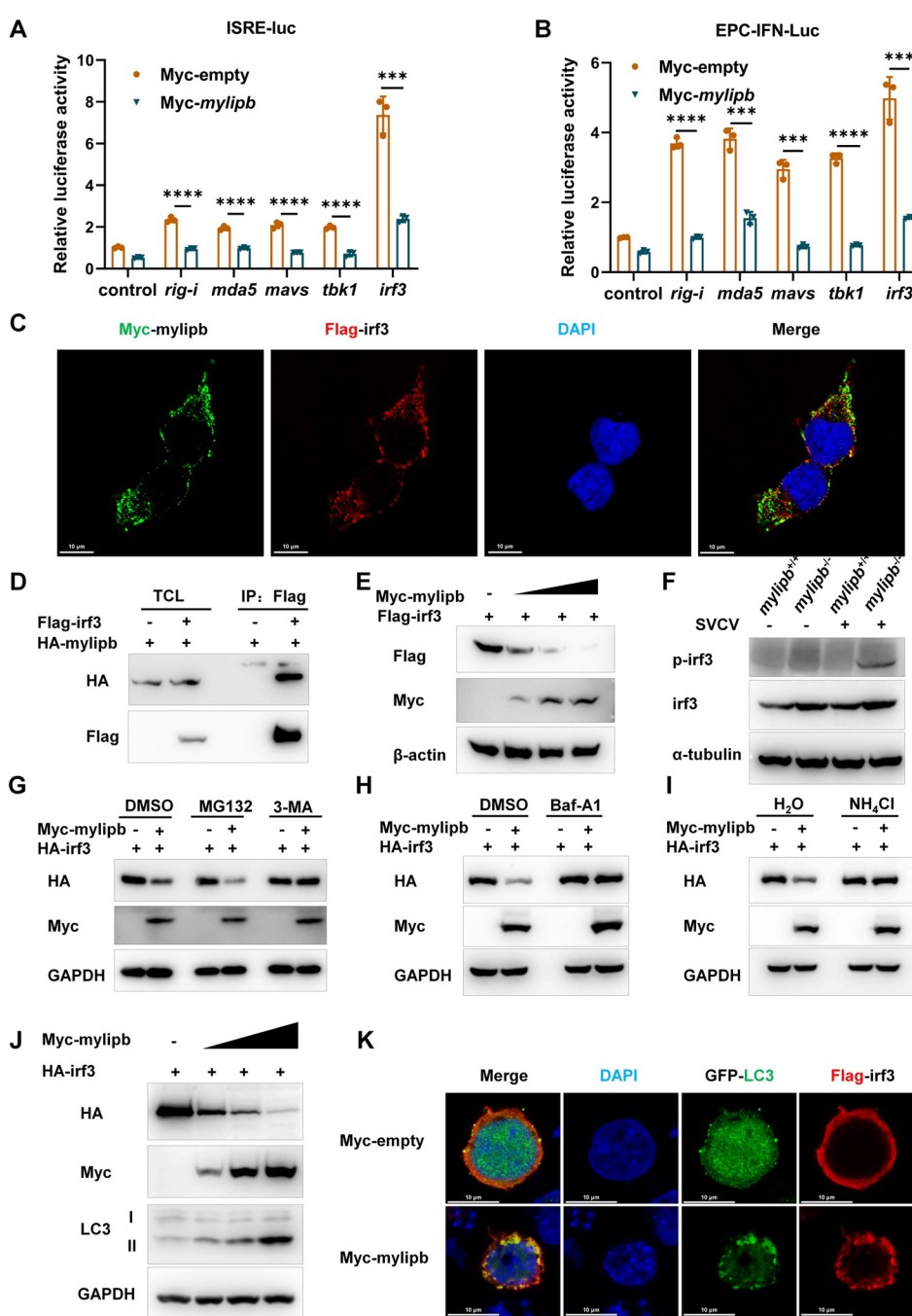

**Fig 4. Mylipb interacts with irf3 and induces autophagic degradation of irf3.** (A) Overexpression of *mylipb* suppressed *rig-i–*, *mda5- mavs-*, *tbk1-*, and *irf3*-induced activation of ISRE luciferase activity in EPC cells. (B) Overexpression of *mylipb* suppressed *rig-i–*, *mda5- mavs-*, *tbk1-*, and *irf3*-induced activation of EPC–IFN promoter luciferase reporter (EPC-IFN-luc) activity in EPC cells. (C) Immunofluorescence staining showed the predominant colocalization of mylipb and irf3 protein in the cytoplasm. HEK293T cells were co-transfected with Myc-*mylipb* and Flag-*irf3*. After 24 h, the cells were fixed and observed by confocal microscopy. Red signals represent overexpressed irf3, green signals represent overexpressed mylipb. (D) Mylipb associated with irf3. HEK293T cells seeded in 100-mm dishes were transfected with the indicated plasmids (4 μg each). After 24 h, total cell lysates were immunoprecipitated (IP) with anti-Flag antibody conjugated agarose beads. Then, the immunoprecipitates and cell lysates were detected with anti-HA or anti-Flag Ab, respectively. (E) Mylipb induced the degradation of irf3 in a dose-dependent manner. HEK293T cells were transfected with Myc-empty, Myc-*mylipb*, and Flag-*irf3* for 24 h, and then the cells were harvested to perform immunoblotting. (F) The protein level of irf3 and p-irf3 was higher in *mylipb*-null zebrfish liver compared with that in WT zebrafish. *mylipb*-null zebrafish (3 mpf) and their WT siblings (3 mpf) were i.p. injected with (+) or

without (-) SVCV at 10 µl/individual for 48 h, then their livers were dissected for Western blot analysis using anti-irf3, anti-p-irf3. (G) Immunoblotting of lysates from HEK293T cells transiently transfected with Myc-empty, Myc-*mylipb*, and HA-*irf3* for 24 h and then cultured in the presence of the proteasome inhibitor, MG132 (20 µM), autophagy inhibitor, 3-MA (1 mM),or DMSO for 12 h. (H) Immunoblotting of lysates from HEK293T cells transiently transfected with Myc-empty, Myc-*mylipb*, and HA-*irf3* for 24 h and then cultured in the presence of the autophagy inhibitor, Baf-A1 (100 nM), or DMSO for 12 h. (I) Immunoblotting of lysates from HEK293T cells transiently transfected with Myc-empty, Myc-*mylipb*, and HA-*irf3* for 24 h and then cultured in the presence of the lysosome inhibitor, NH₄Cl (5 mM), or H₂O for 12 h. (J) Immunoblotting of lysates from HEK293T cells transiently transfected with Myc-empty, Myc-*mylipb*, and HA-*irf3* for 24 h. (K)) HEK293T cells were co-transfected with Myc-*mylipb* or pCMV-Myc plus Flag-*irf3* and GFP-LC3. After 24 h, the cells were fixed and observed by confocal microscopy. Red signals represent overexpressed irf3, green signals represent LC3 positive autophagosome accumulation.

that C387, which corresponds to zebrafish C381, is evolutionarily conserved between human, mouse and zebrafish (S4A Fig). To investigate whether mylipb could enhance irf3 ubiquitination and depend on its enzymatic activity, we transfected *mylipb*-WT and ZF domain deletion *mylipb* (mylipb-ΔZF) together with flag-*irf3* and His-Ub into HEK293T cells. In the presence of mylipb-WT or mylipb-1-416aa, irf3 ubiquitination was dramatically or partly increased (Figs 5C and S5A). However, substitution of the cysteine at position 381 with serine (C381S), deletion of the ZF domain (ΔZF) in mylipb or mylipb-1-279aa, mylipb-1-380aa abrogated irf3 polyubiquitination (Figs 5C and S5A), indicating that E3 ubiquitin ligase activity was required for mylipb-mediated irf3 ubiquitination. Consistently, we further confirmed that the ZF domian and C381 were also required for mylipb-mediated irf3 degradation (Fig 5D and 5E). Meanwhile, upon SVCV infection, ubiquitination of endogenous irf3 was decreased in *mylipb*-null zebrafish (*mylipb⁻/⁻*) compared with WT zebrafish (*mylipb⁺/⁺*) (Fig 5F).

We also investigated which type of polyubiquitin linked to irf3 was increased by mylipb. As shown in Fig 5G and 5H, overexpression of *mylipb* effectively promoted K6-linked polyubiquitination of irf3, but not K11-, K27-, K29-, K33-, K48- and K63-linked polyubiquitination of irf3 (Figs 5G, 5H, and S5B). We then focused on screening which lysine(s) of irf3 were targeted by mylipb. Homology alignment analysis identified seven conservative lysine residues of mylipb from human, mouse and zebrafish (S6A Fig). Furthermore, overexpression of hu*MYLIP* also destabilized huIRF3 (S6B Fig), suggesting that the ubiquitination sites they contain are evolutionarily conserved. Therefore, we constructed seven irf3 mutants irf3(K5R), irf3(K39R), irf3(K76R), irf3(K86R), irf3(K103R), irf3(K352R) and irf3(K401R) in which lysine residues at positions 5, 39, 76, 86, 103, 352 and 401 were specifically replaced by arginine. Ubiquitination assays showed that overexpression of *mylipb* significantly promoted K6 polyubiquitination of irf3(K5R), irf3(K39R), irf3(K76R), irf3(K86R), irf3(K103R) and irf3(K401R), but not irf3(K352R) (Fig 5H). We further confirmed that overexpression of *mylipb* suppressed the ISRE reporter activity induced by irf3(K5R), irf3(K39R), irf3(K76R), irf3(K86R), irf3(K103R) and irf3(K401R), but not irf3(K352R) (Fig 5I). Consistent with this notion, mylipb could not induce the degradation of irf3(K352R) (Fig 5J). These data suggest that zebrafish mylipb induces degradation of irf3 by promoting K6-linked polyubiquitination of irf3 at lysine 352.

To further investigate whether the enzymatic activity of mylipb is involved in the antiviral response. The results of promoter analysis and qRT-PCR analysis showed that mylipb-WT inhibited poly I:C- and SVCV-induced ISRE reporter and EPC-IFN-luc activity and the transcription of antiviral genes (Fig 6A–6D). Notably, although the enzyme-inactivated mylipb-C381S significantly rescued the inhibitory effect of IFN activation compared to mylipb-WT, it still had a partial negative regulatory function (Fig 6A–6D). Consistently, as shown in Fig 6E, 6F and 6G, overexpression of *mylipb*-C381S significantly impaired the ability to increase CPE, viral titer and mRNA expression of viral genes after SVCV infection in EPC cells compared to

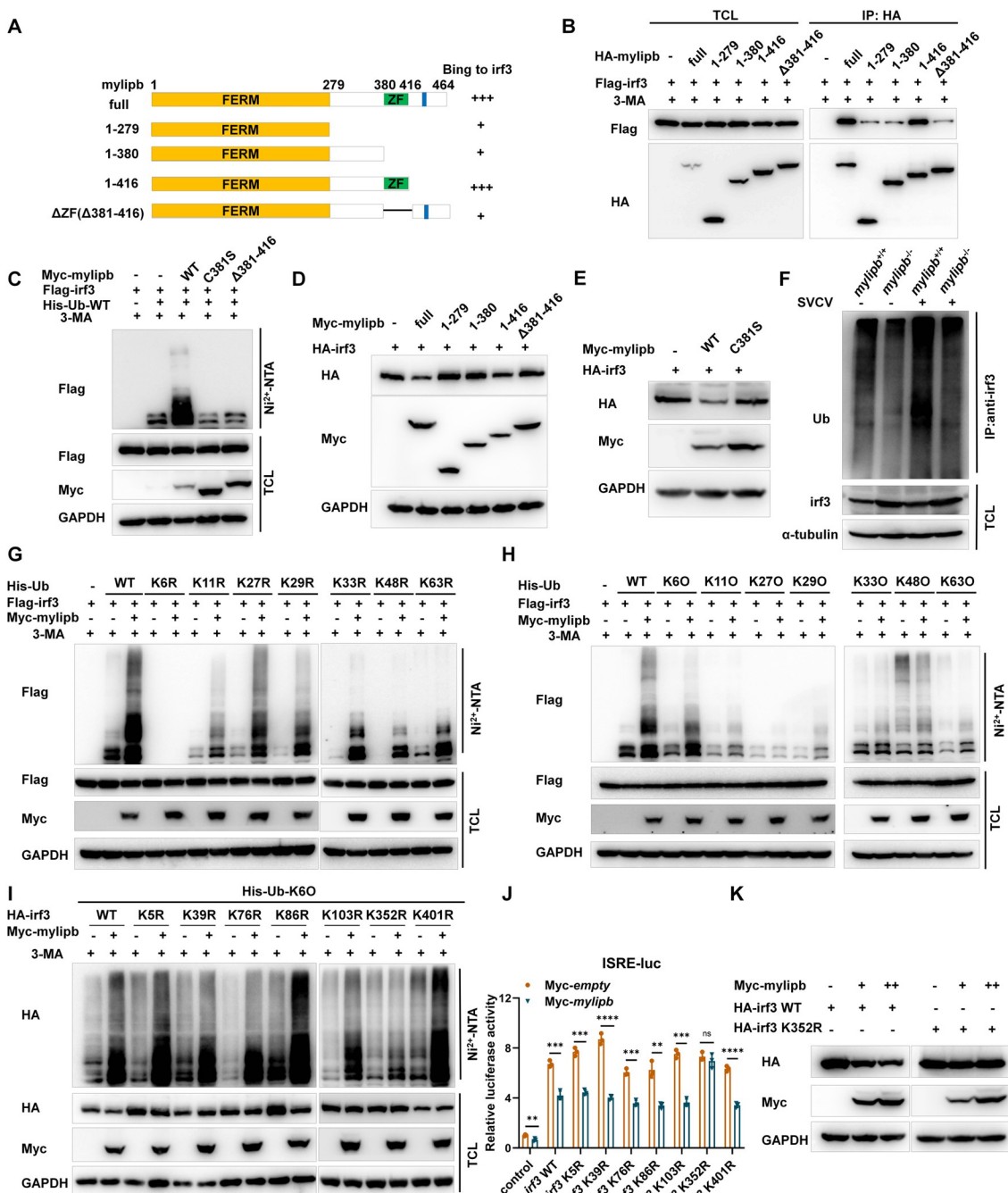

**Fig 5. Mylipb negatively regulates irf3 by promoting K6-linkd ubquitination of irf3 at lysine 352.** (A-B) The interaction between mylipb and irf3 protein mostly depended on the ZF domain of mylipb protein. HEK293T cells seeded in 100-mm dishes were transfected with the indicated plasmids (4 μg each). After 24 h, the cells were treated with 3-MA for 8 h. Total cell lysates were immunoprecipitated (IP) with anti-HA antibody conjugated agarose beads. Then, the immunoprecipitates and cell lysates were detected with anti-HA or anti-Flag Ab, respectively. (C) Mylipb promoted irf3 ubiquitination depended on ZF domain and ubiquitin ligase activity of mylipb. HEK 293T cells were transfected with Flag-*irf3*, Myc-*mylipb*-WT, Myc-*mylipb*-Δ381–416, Myc-*mylipb*-C381S, or empty vector, together with His-ubiquitin. At 24 h posttransfection, the cells were treated with 3-MA for 8 h. The cells were lysed using guanidinium chloride, and purified with Ni$^{2+}$-NTA agarose. (D) Mylipb induced the degradation of irf3 dependent of its ZF domain. HEK293T cells were transfected with Myc-*mylipb*-WT or its mutants, together with HA-*irf3* for 24 h. The lysates were then subjected to IB with the indicated Abs. (E) Mylipb induced the degradation of irf3 dependent of its ubiquitin ligase activity. HEK293T cells were transfected with Myc-*mylipb*-WT or Myc-*mylipb*-C381S, together with HA-*irf3* for 24 h. The lysates were then subjected to IB with the indicated Abs. (F) Ubiquitination assay of irf3 in *mylipb*-null (*mylipb*$^{-/-}$) zebrafish liver and their WT (*mylipb*$^{+/+}$) siblings. *mylipb*-null zebrafish (3 mpf) and their WT siblings (3 mpf) were i.p. injected with (+) or without (-) SVCV

($\sim 2.50 \times 10^7$ TCID$_{50}$/mL) at 10 μl/individual for 48 h, then their livers were dissected, and lysed with lysis buffer (100 μl). The supernatants were denatured at 95°C for 5 min in the presence of 1% SDS. The denatured lysates were diluted with lysis buffer to reduce the concentration of SDS (<0.1%). Denature-IP was conducted using anti-irf3 antibody and then subjected to immunoblotting with anti-Ubiquitin antibody. (G-H) Mylipb promoted K6-linked poly-ubiquitination of irf3. (I) Mylipb promoted K6-linked poly-ubiquitination at lysine 352 of irf3. (G-I) HEK 293T cells were transfected with Flag-*irf3* or HA-*irf3*, Myc-*mylipb*, or empty vector, together with His-ubiquitin or its mutants. At 24 h posttransfection, the cells were treated with 3-MA for 8 h. The cells were lysed using guanidinium chloride, and purified with Ni$^{2+}$-NTA agarose. IB, immunoblot; KO, K-only; KR, K is mutated to R. (J) K352R mutation of irf3 abolished the inhibitory effect of mylipb on ISRE reporter activities in EPC cells. EPC cells were transfected with ISRE reporter and HA-*irf3* or its mutants together with Myc-*mylipb* or empty vector for 24 h, and then conducted luciferase reporter activity assays. (K) K352R mutation abolished degradation of irf3 induced by mylipb. HEK293T cells were transfected with Myc-*mylipb* or empty vector, together with HA-*irf3* or HA-*irf3*-K352R for 24 h. The lysates were then subjected to IB with the indicated Abs.

overexpression of *mylipb*-WT. However, overexpression of enzyme-inactivated mylipb still had a statistically significant enhancing effect compared to empty vector transfection (Fig 6E–6G).

Taken together, the results suggest that mylipb degrades irf3 in a manner dependent on its ubiquitin ligase activity, but also suggest that there are other mechanisms by which mylipb negatively regulates antiviral responses in addition to promoting irf3 degradation.

## Mylipb decreases tbk1-mediated irf3 phosphorylation and cellular antiviral response

Zebrafish tbk1 kinase activity and tbk1-mediated irf3 phosphorylation play an important role in defence against viral infection. To investigate whether mylipb could regulate tbk1-mediated irf3 phosphorylation in the antiviral response, we first overexpressed *tbk1* in the absence or presence of mylipb and examined the gel shift of irf3. As shown in Figs 7A and S7A, overexpression of *tbk1* resulted in a shift of irf3 in the gel, which was reduced by CIP treatment (Figs 7A and S7A). However, in the presence of either mylipb-WT or mylipb-C381S, the shift band became weaker compared to that in the absence of mylipb, suggesting that mylipb could regulate tbk1-mediated irf3 phosphorylation (Fig 7A). Subsequent immunoblot analysis indicated that mylipb did not significantly disrupt the protein level of tbk1, which was distinct from mylipb-mediated degradation of irf3 (Fig 7B). Notably, in the presence of tbk1 did indeed cause a gel shift of mylipb, which was reduced by CIP treatment (Figs 7B and S7B), suggesting that mylipb may be a substrate for the phosphokinase tbk1. To support this speculation, we examined protein-protein interactions. As shown in Fig 7C, coimmunoprecipitation assays revealed that mylipb associated with tbk1(Fig 7C), and overexpression of both HA-*tbk1* and flag-*tbk1* caused a gel shift of mylipb (Fig 7D and 7E). However, the gel shift of mylipb disappeared after overexpression of enzyme-inactivated *tbk1* (tbk1-R47H) [63] (Figs 7F and S8). These results indicated that mylipb was a substrate for the phosphokinase tbk1. To determine the phosphorylation sites of mylipb, we performed a search in the phosphosite database (https://www.phosphosite.org/) and identified 8 conservative points of huMYLIP, including T47, S49, S149, T279, S282, T318, S319 and Y323. By homology alignment analysis, the corresponding zebrafish mylipb for these sites are T47, T49, S149, T279, S282, T318, S319 and Y323 (S9A Fig). Since huMYLIP can also be phosphorylated by huTBK1 (S9B Fig), we speculate that the phosphorylation sites they contain are evolutionarily conserved. Given that tbk1 is a serine/threonine phosphokinase, the following mutant plasmids including T47/S49A(mut1), S149A(mut2), T279/S282A(mut3) and T318/S319A(mut4) were constructed for further experiments. As shown in Fig 7G, overexpression of *tbk1* caused hardly any gel shift of mylipb-mut4, suggesting that T318/S319 are the phosphorylation sites of mylipb catalyzed by tbk1. Based on these data, it is possible that the host uses mylipb to compete with cellular irf3 for phosphorylation by tbk1. As expected, immunoblot analysis showed that the presence of

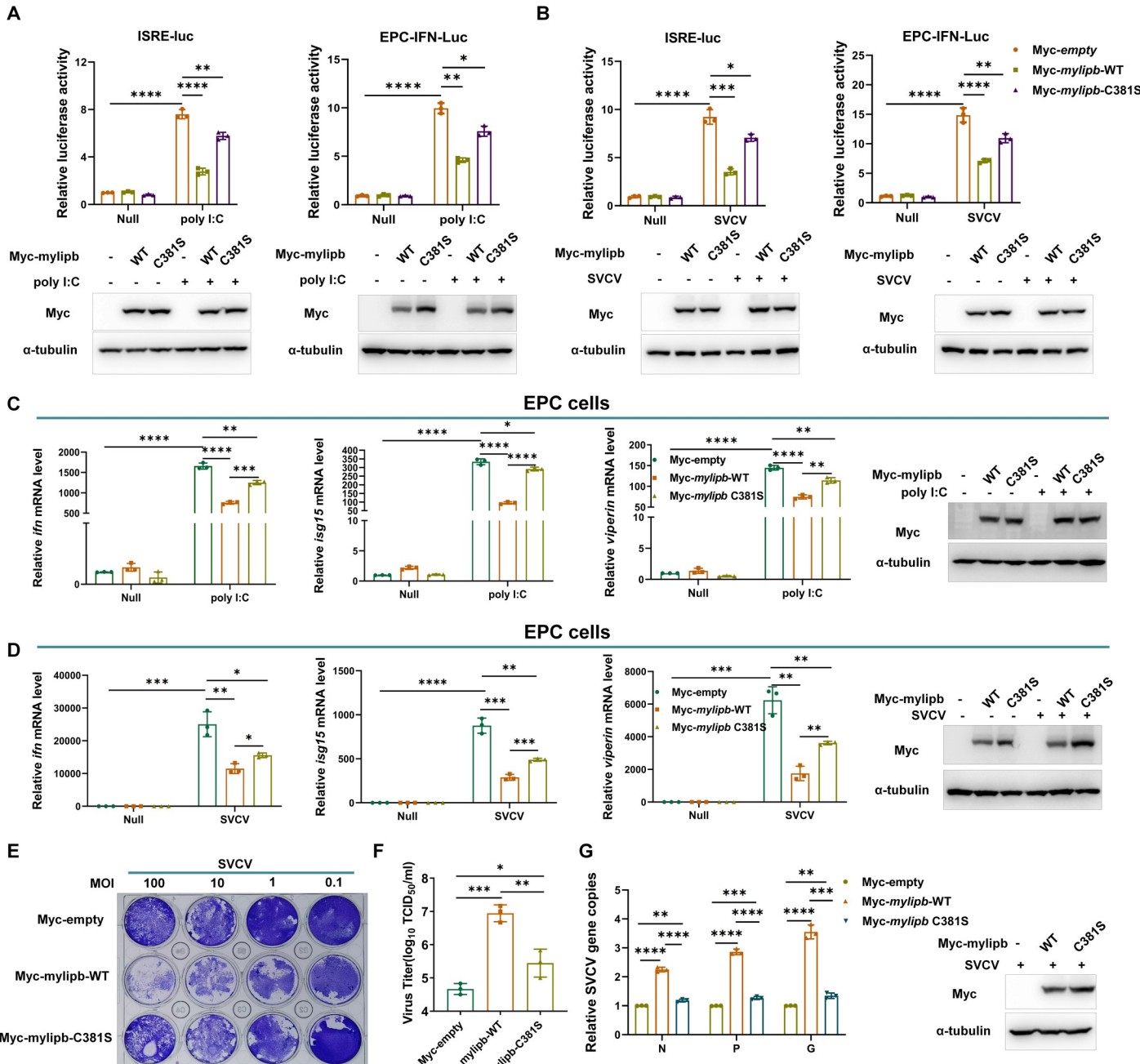

**Fig 6. Mylipb negatively regulates cellular antiviral response mainly dependent of its ubiquitin ligase activity.** (A) ISRE reporter activity and EPC-IFN-Luc activity in Myc-empty vector or Myc-*mylipb*-WT and Myc-*mylpib*-C381S-transfected EPC cells for 24 h, followed by transfection with or without poly I:C (1μg/mL) for 24 h. Western blotting tests to detect the overexpression of *mylipb*-WT *and mylipb*-C381S. (B) ISRE reporter activity and EPC-IFN-Luc activity in Myc-empty vector or Myc-*mylipb*-WT and Myc-*mylipb*-C381S-transfected EPC cells with or without SVCV (MOI of 1) infection for 24 h. Western blotting tests to detect the overexpression of *mylipb*-WT *and mylipb*-C381S. (C) qRT-PCR of *ifn*, *isg15*, and *viperin* mRNA in EPC cells transfected with Myc-empty or or Myc-*mylipb*-WT and Myc-*mylipb*-C381S-transfected EPC cells for 24 h, followed by transfection with or without poly I:C (1μg/mL) for 24 h. Western blotting tests to detect the overexpression of *mylipb*-WT *and mylipb*-C381S. (D) qRT-PCR of *ifn*, *isg15*, and *viperin* mRNA in EPC cells transfected with Myc-empty or or Myc-*mylipb*-WT and Myc-*mylipb*-C381S-transfected EPC cells for 24 h, followed by infection with or without SVCV (MOI of 1) for 24 h. Western blotting tests to detect the overexpression of *mylipb*-WT *and mylipb*-C381S. (E) The plaque assay of EPC cells transfected with Myc-empty or or Myc-*mylipb*-WT and Myc-*mylipb*-C381S-transfected EPC cells for 24 h, followed by infection with or without SVCV(MOI:0.1–100) for 48 h. (F) Determination of SVCV titre in culture medium of EPC cells transfected with Myc-empty or or Myc-*mylipb*-WT and Myc-*mylipb*-C381S-transfected EPC cells for 24 h, followed by infection with or without SVCV(MOI:1) for 48 h. (G) qRT-PCR of *N*, *P*, and *G* mRNA in EPC cells transfected with Myc-empty or or Myc-*mylipb*-WT and Myc-*mylipb*-C381S-transfected EPC cells for 24 h, followed by infection with or without SVCV for 24 h. Western blotting tests to detect the overexpression of *mylipb*-WT *and mylipb*-C381S. qRT-PCR data are presented as mean values based on three repeated experiments, and error bars indicate the ± SD. *,$P < 0.05$, **, $P<0.01$; ***, $P < 0.001$; ****,$P< 0.0001$.

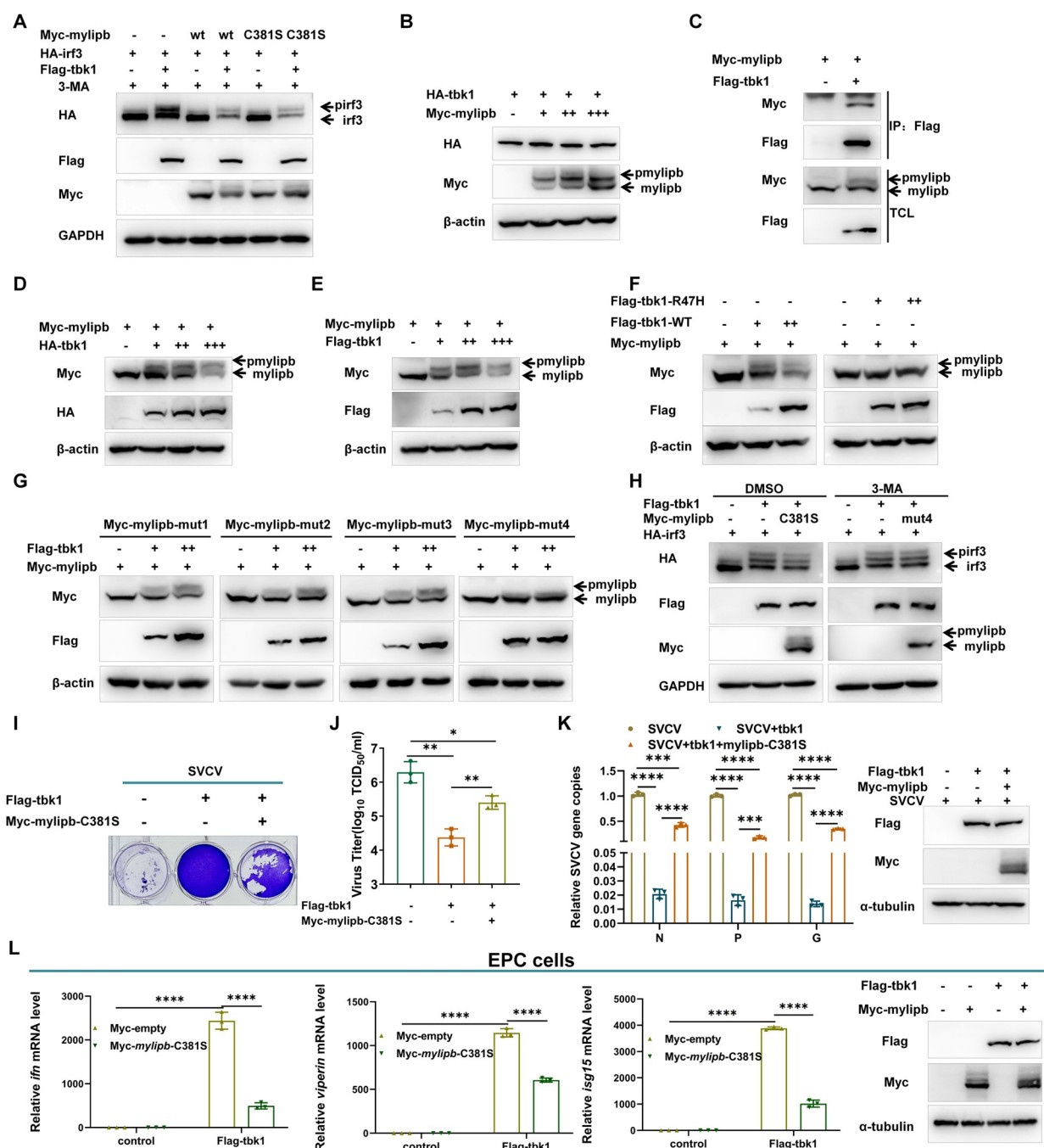

**Fig 7. Mylipb decreases tbk1-mediated irf3 phosphorylation and cellular antiviral response.** (A) Overexpression of *mylipb*-WT and *mylipb*-C381S inhibits tbk1-mediated phosphorylation of irf3. HEK293T cells were transfected with Flag-*tbk1* and empty vector or Myc-*mylipb*-WT or *mylipb*-C381S, together with HA-*irf3* for 24 h. At 24 h posttransfection, the cells were treated with 3-MA for 8 h. The lysates were then subjected to IB with the indicated Abs. (B) Mylipb interacts with tbk1. HEK293T cells seeded in 100-mm dishes were transfected with the indicated plasmids (4 μg each). After 24 h, total cell lysates were immunoprecipitated (IP) with anti-Flag antibody conjugated agarose beads. Then, the immunoprecipitates and cell lysates were detected with anti-Myc or anti-Flag Ab, respectively. (C) Overexpression of *mylipb* does not significantly disrupt the protein level of tbk1 in dose manner. (D-E) Tbk1 phosphorylates mylipb. HEK293T cells co-transfected with HA-*tbk1* or Flag-*tbk1* and empty vector together with Myc-*mylipb* for 24 h. The lysates were then subjected to IB with the indicated Abs. (F) Tbk1 phosphorylates mylipb dependent on its phosphokinase activity. HEK293T cells were transfected with Flag-*tbk1* or Flag-*tbk1*-R47H (enzyme inactivated mutation) and empty vector together with Myc-*mylipb* for 24 h. The lysates were then subjected to IB with the indicated Abs. (G) T318/S319 is the phosphorylation sites of mylipb catalyzed by tbk1. HEK293T cells were transfected with Myc-*mylipb* mutants together with Flag-*tbk1* or empty vector for 24 h. The lysates were then subjected to IB with the indicated Abs. (H) Mylipb-mut4(T318/S319A) has no obvious effect on phosphorylation of irf3. HEK293T cells were transfected with Flag-*tbk1* and empty vector or Myc-*mylipb*-C381S or *mylipb*-mut4,

together with HA-*irf3* for 24 h, and then cultured in the presence of 3-MA (1 mM),or DMSO for 12 h. The lysates were then subjected to IB with the indicated Abs. (I-J) Overexpression of *mylipb* decreases tbk1-mediated decline of viral titer. EPC cells seeded in 12-well plates overnight were transfected with Flag-*tbk1* and Myc-*mylipb*-C381S or empty vector. At 24 h post-transfection, cells were infected with SVCV (MOI = 1) for 48 h. Then, cells were fixed with 4% PFA and stained with 1% crystal violet (I). Culture supernatants from the cells infected with SVCV were collected, and the viral titer was measured (J). (K) Overexpression of *mylipb* decreases tbk1-mediated decline of copy number of SVCV genes in SVCV-infected EPC cells. EPC cells were transfected with Flag-*tbk1* and Myc-*mylipb*-C381S or empty vector, cells were infected with SVCV (MOI of 1). After 24 h, total RNAs were extracted for examining the mRNA levels of the *N*, *P*, and *G* gene of SVCV by qRT-PCR analysis. Western blotting tests to detect the overexpression of *mylipb* and *tbk1*. (L) Overexpression of *mylipb* blocks the expression of *ifn*, *viperin* and *isg15* induced by tbk1. EPC cells were transfected with Flag-*tbk1* or empty vector together with Myc-*mylipb*-C381S or empty vector. At 24 h after transfection, total RNAs were extracted for examining the mRNA levels of the *ifn*, *viperin*, and *isg15* gene of EPC cells by qRT-PCR analysis. Western blotting tests to detect the overexpression of *mylipb* and *tbk1*. All data are presented as mean values based on three repeated experiments, and error bars indicate the ± SD. *,$P < 0.05$, **, $P<0.01$; ***, $P < 0.001$; ****, $P< 0.0001$.

mylipb-C381S attenuated the shifted band of irf3, but mylipb-mut4 had no apparent effect on reducing the phosphorylation of irf3 (Fig 7H). Given that mylipb is a decoy substrate for tbk1 and that tbk1 plays a critical role in the defence against viral infection, the function of mylipb in regulating the tbk1-mediated cellular antiviral immune response was investigated. Upon SVCV infection, zebrafish mylipb-C381S interfered with the antiviral function of tbk1 in terms of both CPE and viral titer (Fig 7I–7J). In addition, the effect of tbk1 on viral gene inhibition was attenuated by mylipb (Fig 7K). For the host antiviral IFN response, the tbk1-induced upregulation of *ifn* and ISGs was also reversed by mylipb (Fig 7L).

These results suggest that mylipb acts as a decoy substrate to attenuate the cellular antiviral response capacity induced by phosphokinase tbk1-mediated irf3 phosphorylation.

## Mylipb interacts with irf7 and induces degradation of irf7

We further test if mylipb also targets irf7 and NF-κB proteins. Co-immunoprecipitation assays demonstrated that ectopically expressed mylipb pulled down irf7 but not NF-κB–p65 in HEK293T cells (S10A and S10B Fig). Overexpression of mylipb resulted in a dose-dependent reduction of overexpressed irf7 protein (S10C Fig). However, we found that irf7 degradation induced by mylipb was detectable in both wild-type (WT) cells and ATG5 KO or BECN1 KO cells (S10D Fig), suggesting that mylipb induces degradation of irf7 independent of the autophagy pathway. As shown in S10E Fig, mylipb induced the degradation of irf7 dependent of its ubiquitin ligase activity (S10E Fig). Using promoter assays, we found that overexpression of mylipb-WT suppressed the ISRE and IFN reporter induced by irf7(S10F–S10G Fig). These results suggest that mylipb also targets irf7, but is mechanistically distinct from irf3. We will investigate this further at a later stage.

## Discussion

IRF3 is a transcription factor essential for innate immunity against viruses, and its protein levels and activity can be precisely regulated by post-translational modifications to effectively protect the host from infection and prevent excessive immune pathology [15,21]. In this study, we identified zebrafish *mylipb* as a virus-responsive gene that negatively regulates antiviral innate immunity by promoting K6-linked polyubiquitination of irf3 at lysine 352 and inhibiting tbk1-mediated phosphorylation of irf3, uncovering a novel, to our knowledge, function of *mylipb* of innate antiviral responses using an in *vivo* model (Fig 8).

Ubiquitination is a post-translational modification with complex modes and diverse results [34]. It plays an important role in the vital activities by regulating the stability, activity and localization of target proteins [64]. The ubiquitin-mediated system is one of the major degradation mechanisms for the removal of abnormal and unused proteins in order to maintain intracellular protein homeostasis [65]. In addition to the classical K48-linked

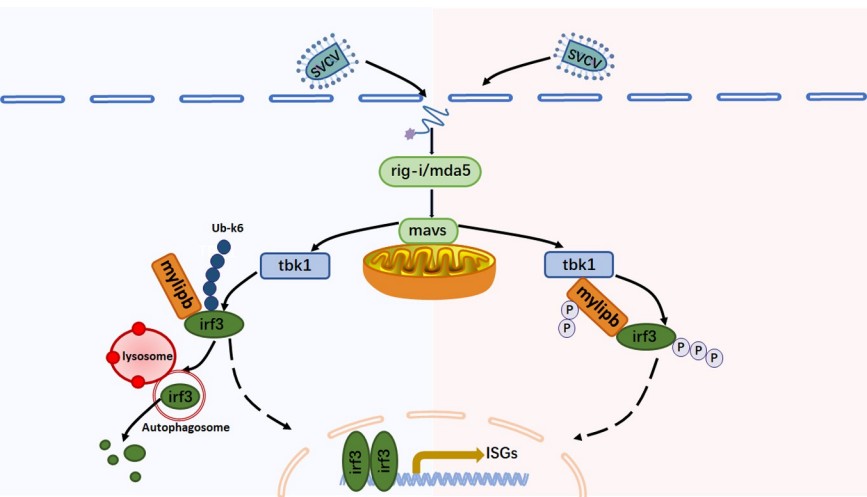

**Fig 8. A model on the two synergistic mechanisms of zebrafish mylipb in negative regulation of antiviral innate immunity.** Upon viral infection, zebrafish rig-i/mda5 recognizes the viral RNA and interacts with mavs, leading to the recruitment of tbk1, which phosphorylates irf3 and then induces IFN production. Zebrafish mylipb, an E3 ubiquitin ligase, increases K6-linked polyubiquitination at Lys 352, leading to the induction of autophagic degradation of irf3, while at the same time reducing irf3 phosphorylation by acting as a substrate for tbk1, thereby preventing the expression of IFN.

polyubiquitination, all non-K63-linked (including K6-, K11-, K27-, K29- and K33-linked) ubiquitination could result in proteasomal degradation of proteins [66]. Additionally, increasing evidence has shown that the ubiquitination proteins can also be degraded by the lysosomal and autophagy pathways [40,67,68]. Previous studies have shown that different ubiquitination events are involved in the regulation of IRF3 stability. RBCK1, c-Cbl and RAUL induces K48-linked polyubiquitination and degradation of IRF3 by the proteasomes, thereby repressing the expression of type I IFNs [43,44,46]. Zebrafish fbxo3 increases the K27-linked polyubiquitination of irf3 and promotes its degradation by the proteasome pathway [42]. TRIM21 and PMSD14 regulate the autophagy of IRF3 by balancing the K27-linked polyubiquitination of IRF3 [40,69]. Furthermore, even the same E3 ubiquitin ligase causes substrates to undergo different degradation pathways due to the diversity of ubiquitination types. For example, as an E3 ubiquitin ligase, MYLIP inhibits LDLR and VLDRL proteins via the ubiquitin-lysosomal pathway, while controlling its own protein levels via the ubiquitin-proteasome pathway [54, 70]. This study reports that zebrafish mylipb interacts with irf3 and catalyses the K6-linked polyubiquitination of irf3, dependent on its E3 ubiquitin ligase activity, leading to the degradation of irf3 by the autophagy pathway.

Protein undergoing K6-linked ubiquitination, resulting in different propensities to be stabilised, degraded, change in cellular localisation and ability to bind partners, etc [71,72]. The role of K6-linked ubiquitination in antiviral innate immunity is attracting the attention of researchers. It is a degradation signal for viperin and IRF7, and regulates the DNA-binding capacity of IRF3 [38,73,74]. Here, we report that K6-linked ubiquitination is an autophagy signal for irf3, revealing the functional multiplexity of K6-linked ubiquitination. The function of K6-linked ubiquitination in innate immunity has been established, but the diverse patterns of regulation and the molecular mechanisms involved remain to be elucidated.

Upon pathogen recognition, IRF3 is activated through a sequential process including phosphorylation, dimerization and nuclear translocation [21,22]. Thus, modulation of IRF3 phosphorylation usually occurs in the cytoplasm, mediated by disruption of TBK1 function [18,47].

In this study, we show that cytoplasmic mylipb associates with tbk1 and acts as a decoy substrate to attenuate tbk1-mediated irf3 phosphorylation, thereby impairing the cellular antiviral response capacity. Our data provide a synergistic mechanism for mylipb in antiviral immunity independent of its enzymatic activity. However, we do not provide sufficient evidence to suggest that zebrafish mylipb could also reduce the dimerization and nuclear translocation of irf3 after infection with SVCV.

Viral infections in fish cause huge losses in the aquaculture industry [75–77]. In this study, we find that *mylipb*-deficient zebrafish are more resistant to SVCV infection, but have no defects in development, growth rate and reproductive ability. These data suggest that *mylipb* may be a good candidate gene for the cultivation of antiviral fish strains using gene-targeting technology, which would greatly benefit the aquaculture industry.

In summary, we have mainly used the zebrafish model and cell line to demonstrate the function of mylipb acting as a suppressor in the antiviral innate immunity. Given the evolutionarily conserved activity of MYLIP between zebrafish and mammals, mammalian *MYLIP* may have similar function as zebrafish *mylipb* in response to viral infection.

## Materials and methods

### Ethics statement

Fish experiments in this study were conducted at the Institute of Hydrobiology, Chinese Academy of Sciences, in accordance with European Union guidelines for handling of laboratory animals (2010/63/EU). All zebrafish experiments were approved by the Institutional Animal Care and Use Committee of the Institute of Hydrobiology, Chinese Academy of Sciences (IHB-2022-039).

### Cells and viruses

Zebrafish liver cells (ZFL) were grown in Ham's F12 medium (HyClone) supplemented with 10% FBS. Epithelioma papulosum cyprini (EPC) cells were maintained in medium 199(M-199) (ViaCell Biosciences) containing 10% FBS. HEK 293T cells were maintained at 37˚C containing 5% CO2, and ZFL and EPC cells were grown at 28˚C containing 5% CO2. Human embryonic kidney (HEK) 293T cells were cultured in Dulbecco's modified Eagle medium (DMEM) (ViaCell Biosciences) supplemented with 10% fetal bovine serum (FBS) (ViaCell Biosciences). All cell lines were cultured in a humidified incubator, verified to be free of Mycoplasma contamination before use. Spring viremia of carp virus (SVCV), strain OMG067 was propagated in EPC cells until the cytopathic effect (CPE) was complete, and the culture medium containing SVCV (multiplicity of infection [MOI]; $10^9$ $TCID_{50}$/mL) was collected and stored at -80˚C until use.

### Zebrafish

Zebrafish (Danio rerio) strain AB fish were raised, maintained, and staged according to standard protocols. CRISPR/Cas9 was used to knockout *mylipb* in zebrafish. First, *mylipb* single guide RNA (sgRNA) was designed using the CRISPR design tool (http://crispr.mit.edu). The sgRNA sequence was AGATCTACGGTCAAGACCCC. The zebrafish codon optimized Cas9 plasmid was digested with XbaI and then purified and transcribed using the T7 mMessage machine kit (Ambion). PUC9 guide RNA (gRNA) vector was used to amplify *mylipb* sgRNA template. The Transcript Aid T7 highyield transcription kit (Fermentas) was used to synthesize sgRNA. One-cell stage embryos were injected with 1ng Cas9 RNA and 0.15ng sgRNA per embryo. The chimeras were initially detected using a heteroduplex mobility assay (HMA) as

previously described[78].If the HMA results were positive, the remaining embryos were raised to adulthood as the F0 generation and then backcrossed to WT zebrafish (strain AB) to generate the F1 generation. F1s were genotyped using HMAs. Genotypes were confirmed by target site sequencing. Heterozygous F1s were backcrossed to WT zebrafish (strain AB; parent-offspring mating prohibited) to generate the F2 generation. F2 adults carrying the target mutation were intercrossed to generate F3 offspring. The F3 generation contained WT (+/+), heterozygous (+/-), and homozygous (-/-) individuals. The primers used to identify mutants were as follows: forward primer,CTGATGCATGTGAAGGAGGA, reversed primer,GTAACG-CAACTGATGCTAGG The novel mutants were named following zebrafish nomenclature guidelines: $mylipb^{ihbmb73/\ ihbmb73}$(https://zfin.org/ZDB-ALT-230809-2). All zebrafish procedures were performed approval of the Institutional Animal Care and Use Committee (IACUC) at the Institute of Hydrobiology, Chinese Academy of Sciences.

## Plasmid construction and reagents

Zebrafish gene *mylipb* (ENSDARG00000055118) and its truncated mutants were PCR amplified from AB zebrafish cDNA using PCR. Amplified genes were subcloned into pCMV-Myc (Clontech), pCMV-HA (Clontech), or pCMV-Flag (Clontech) vectors. These plasmids, including Myc-*rig-i*, Myc-*mda5*, Myc-*tbk1*, Myc-*irf3*, were constructed using the pCMV-Myc vector as described previously. All constructs were verified by DNA sequencing.

VigoFect (Vigorous Biotechnology, Beijing, China) and FishTrans (MST, FT2020, Wuhan, China) were used for cell transfection. poly I:C (InvivoGen, San Diego, CA) was transfected with Lipofectamine 3000 (Thermo Fisher Scientific). Cycloheximide (CHX), MG132, Baf-A1 and 3-MA were purchased from MedChem Express. $NH_4Cl$ was purchased from Sigma-Aldrich. The antibodies used were as follows: anti-Flag antibody (F1804; Sigma-Aldrich), anti-Myc antibody (9E10; Santa Cruz Biotechnology), anti-HA antibody (MMS-101R; Covance), anti-LC3 antibody (ab48394; Abcam), anti-irf3 (A11921, Abclonal), anti-phosphor-IRF3 (4947, Cell Signaling Technology). anti-ATG5 (A0203, ABclonal), anti-BECN1 (A17028, ABclonal), anti-α-tubulin (62204, Thermo Fisher Scientific), anti-GAPDH antibody (AC033; ABclonal) and anti-β-actin antibody (AC026; ABclonal).

## Virus infection and plaque assay

For examining gene expression of zebrafish larvae, 30 larvae (3 dpf) were pooled in a disposable 60-mm cell culture dish filled with 5 mL of egg water and 0.5 mL of SVCV ($\sim 2.5 \times 10^7$TCID$_{50}$/ml) solution at 28°C for 24 h. The total RNA was extracted, and quantitative real-time PCR (qRT-PCR) assays were conducted. For survival ratio assays, zebrafish larvae were placed in 96-well plates individually, and then 100μL of egg water containing SVCV (5 mL of egg water plus 2 mL of SVCV ($\sim 2.5 \times 10^7$TCID$_{50}$/ml) was added to each well). The survival ratio was monitored every 1 h over a 24 h period. For viral injection of adult zebrafish, zebrafish (3mpf) were intraperitoneally (i.p.) injected with SVCV($\sim 2.5 \times 10^7$TCID$_{50}$/ml) at 10 μL per individual. Zebrafish i.p. injection with M-199 cell culture medium was used as the control. The survival ratio was monitored every 12 h over a 108 h period.

For the plaque assay, EPC cells were transfected with 2 μg of Myc- *mylipb* or pCMV-Myc vector. After 24 h, cells were infected with SVCV for 48 h at the dose indicated. Subsequently, the supernatant was collected from EPC cells infected with SVCV (multiplicity of infection [MOI] of 1) for the detection of virus titers, and the cells were washed with phosphate-buffered saline (PBS), fixed with 4% paraformaldehyde, and stained with 1% crystal violet for visualizing CPE.

## Virus titer determination

EPC cells were cultured in 96-well plates. The culture supernatant (containing virus) was diluted in serial dilutions ($10^{-1}$ to $10^{-11}$) in sterile 1.5 mL tubes using M-199 medium. Subsequently, the diluted viruses were added to EPC cells seeded into 96-well plates. After 4 days, the plates were observed under a microscope, and the area of detached cells after infection with virus. 50% in one well were counted as positive. The titers for SVCV infection were calculated using the Spearman-Kärber method and represented as 50% tissue culture-infective dose (TCID50) assay. The experiments were repeated three times for statistical analysis.

## Luciferase reporter assays

EPC cells were grown in 24-well plates and transfected with the indicated plasmids, as well as with CMV-Renilla used as a control. The ratio between the luciferase reporters (ISRE-Luc, Dr-IFNφ1-Luc, or EPC-IFN-Luc) and CMV-Renilla was 10:1, and pCMV-Myc vector was used to ensure equivalent amounts of total DNA in each well. After the cells were transfected for 24–48 h, the luciferase activity was determined by the dual-luciferase reporter assay system (Promega). Data were normalized to Renilla luciferase. Data are representative of three independent experiments (mean ± SD).

## Quantitative real-time PCR (qRT-PCR) analysis

Total RNA was extracted from cells, larvae (n = 30), using the RNAiso Plus kit (TaKaRa Bio, Beijing, China), following the manufacturer's instructions. Equivalent amounts of total RNA (2ug) were used for cDNA synthesis with the RevertAid first-strand cDNA synthesis kit (Thermo Scientific, Waltham, MA, USA). MonAmp SYBR green qPCR mix (high Rox) (Monad Bio, Shanghai, China) was used for qRT-PCR assays. Each experiment was repeated at least three times independently. The primers for qRT-PCR assays are listed in S1 Table.

## Immunoprecipitation assay and Western blot

Anti-Flag antibody, anti-Myc antibody and anti-HA antibody-conjugated agarose beads were purchased from Sigma-Aldrich. The experimental procedures including Western blot and coimmunoprecipitation analysis have been described previously [79]. The Fuji Film LAS4000 mini luminescent image analyzer was used for photographing the blots. Multigauge V3.0 was used for quantifying the protein levels based on the band density obtained in the Western blot analysis.

## Immunofluorescence confocal microscopy

HEK293T cells grown on glass coverslips were fixed with 4% paraformaldehyde for 15 min, permeabilized with 0.1% Triton X-100, and blocked with 1% bovine serum albumin (BSA). Then, the cells were stained with the indicated primary antibodies followed by incubation with fluorescent-dye-conjugated secondary antibodies. Nuclei were counterstained with DAPI (4,6-diamidino-2-phenylindole; Sigma-Aldrich). Imaging of the cells was carried out using a Leica laser-scanning confocal microscope.

## Ubiquitination assay

HEK 293T cells seeded in 100-mm dishes were transfected with Flag-*irf3* or HA-*irf3*, Myc-*mylipb*, or empty vector, together with His-ubiquitin or its mutants. At 24 h posttransfection, the cells were treated with 3-MA for 8 h. Transfected cells were washed twice with 2 ml ice-cold PBS and then digested with 1 ml 0.25% trypsin-EDTA for 3 min until the cells were

dislodged. Cells were resuspended into 1.5 ml EP tube with 1ml PBS, centrifuged at 1000rpm for 3 min, and then the supernatant was removed. The cells were lysed using guanidinium chloride, and purified with $Ni^{2+}$-NTA agarose overnight at 4°C with constant agitation. These samples were further analyzed by immunoblotting. Ubiquitination assay of irf3 in zebrafish liver: *mylipb*-null zebrafish (3 mpf) and their WT siblings (3 mpf) were i.p. injected with (+) or without (-) SVCV($\sim 2.50 \times 10^7$ $TCID_{50}$/mL) at 10 μl/individual for 48 h, then their livers were dissected, and lysed with lysis buffer (100 μl). The supernatants were denatured at 95°C for 5 min in the presence of 1% SDS. The denatured lysates were diluted with lysis buffer to reduce the concentration of SDS (<0.1%). Denature-IP was conducted using anti-irf3 antibody and then subjected to immunoblotting with anti-Ubiquitin antibody.

## Statistical analysis

The statistical analysis was performed using GraphPad Prism version 8.0 (unpaired t tests). Data are representative of at least three independent experiments, and error bars indicate the mean with SD. For zebrafish survival analysis, the Kaplan-Meier method was adopted to generate graphs, and the survival curves were analyzed by log-rank analysis. A *p* value <0.05 was considered significant. Statistical significance is represented as follows: *$p$<0.05; **$p$<0.01; ***$p$<0.001; ****$p$< 0.0001.

## Supporting information

**S1 Fig. The generation of mylipb-null zebrafish via CRISPR/Cas9 techniques.** (A) The schematic of the targeting site in mylipb and the resulting nucleotide sequence in the mutant (MT, *mylipb*[ihbmb73/ ihbmb73]).The predicted protein product of maoc1 in the mutant and wild-type (WT) sibling. (B) Verification of the efficiency of CRISPR/Cas9-mediated zebrafish mylipb disruption by heteroduplex mobility assay (HMA). (C) The relative mRNA levels of mylipb in the WT and homozygous mutant. All data are presented as mean values based on three repeated experiments, and error bars indicate the ± SD. *,$P < 0.05$, **, $P<0.01$; ***, $P < 0.001$; ****, $P< 0.0001$
(TIF)

**S2 Fig. Mylipb induces inflammation factor expression in SVCV infection.** (A) Induction of genes downstream of NF-kB, including *il-1β*, *il8*, *tnfα*, was decreased in mylipb[-/-] larvae compared with WT larvae (*mylipb*[+/+]) after SVCV infection. 30 larvae (3 dpf) were pooled in a disposable 60-mm cell culture dish filled with 5 mL of egg water and 0.5 mL of SVCV ($\sim 2.5 \times 10^7$ $TCID_{50}$/ml) solution at 28°C for 24h. (B) Overexpression of mylipb suppressed expression of *il-1β*, *il8* and *tnfα* induced by SVCV infection in EPC cells. EPC cells were transfected with Myc-mylipb or empty vector for 24 h and infected with SVCV (MOI: 1). After 24 h, total RNAs were extracted for examining the mRNA levels of *il-1β*, *il8* and *tnfα* by qRT-PCR analysis.
(TIF)

**S3 Fig. Mylipb interacts with irf3 and induces autophagic degradation of irf3.** Related to Fig 4.(A) Mylipb associated with irf3. HEK293T cells seeded in 100-mm dishes were transfected with the indicated plasmids (4 μg each). After 24 h, total cell lysates were immunoprecipitated (IP) with anti-HA antibody conjugated agarose beads. Then, the immunoprecipitates and cell lysates were detected with anti-HA or anti-Flag Ab, respectively. (B, C) Wild-type (WT), ATG5 and BECN1 knockout (KO) 293T cells were co-transfected with Myc-mylipb and HA-irf3 for 24 h. The cell lysates were subjected to western blotting with the indicated antibodies.
(TIF)

**S4 Fig.** (A) Amino acid sequence alignment of human MYLIP and zebrafish mylipb.
(TIF)

**S5 Fig. Mylipb negatively regulates irf3 by promoting K6-linkd ubquitination of irf3.**
Related to Fig 5. (A) Mylipb promoted irf3 ubiquitination depended on ZF domain and ubi-
quitin ligase activity of mylipb. HEK 293T cells were transfected with Flag-*irf3*, Myc-*mylipb*-
WT, Myc-*mylipb*-1-279, Myc-*mylipb*-1-380, Myc-*mylipb*-1-416, Myc-*mylipb*-Δ381–416, or
empty vector, together with His-ubiquitin. At 24 h posttransfection, the cells were treated with
3-MA for 8 h. The cells were lysed using guanidinium chloride, and purified with $Ni^{2+}$-NTA
agarose. (B) Mylipb promoted irf3 K6-linked ubiquitination. HEK293T cells were transfected
with Flag-*irf3*, Myc-*mylipb*, or empty vector, together with His-K6R ubiquitin. At 24 h post-
transfection, the cells were treated with 3-MA for 8 h. The cells were lysed using guanidinium
chloride, and purified with $Ni^{2+}$-NTA agarose.
(TIF)

**S6 Fig.** (A) Amino acid sequence alignment of human IRF3, mouse Irf3, zebrafish irf3. (B)
MYLIP induced degradation of IRF3. HEK293T cells co-transfected with Myc-*IRF3* and
empty vector together with HA-*IRF3* for 24 h. The lysates were then subjected to IB with the
indicated Abs.
(TIF)

**S7 Fig. Mylipb decreases tbk1-mediated irf3 phosphorylation.** Related to Fig 7. (A-B) The
amount of tbk1-phosphorylated irf3(A) and mylipb(B) were reduced by CIP treatment. Cells
were transfected with the indicated plasmids (1 μg each) for 24 h. Then the cell lysates (100 μL)
were treated with or without CIP (10 U) for 30 min at 37°C. Then the lysates were detected by
IB with the indicated Abs.
(TIF)

**S8 Fig.** (A) Homology alignment analysis show that the corresponding zebrafish tbk1 for R47
of human TBK1 is R47.
(TIF)

**S9 Fig.** (A) Amino acid sequence alignment of human MYLIP, mouse Mylip, zebrafish
mylipb. (B) TBK1 phosphorylates MYLIP. HEK293T cells co-transfected with HA-*TBK1* and
empty vector together with Myc-*MYLIP* for 24 h. The lysates were then subjected to IB with
the indicated Abs.
(TIF)

**S10 Fig. Mylipb interacts with irf7 and induces degradation of irf7.** (A) Mylipb don't associ-
ated with p65. HEK293T cells seeded in 100-mm dishes were transfected with the indicated
plasmids (4 μg each). After 24 h, total cell lysates were immunoprecipitated (IP) with anti-Flag
antibody conjugated agarose beads. Then, the immunoprecipitates and cell lysates were
detected with anti-Myc or anti-Flag Ab, respectively. (B) Mylipb associated with irf7.
HEK293T cells seeded in 100-mm dishes were transfected with the indicated plasmids (4 μg
each). After 24 h, total cell lysates were immunoprecipitated (IP) with anti-Flag antibody con-
jugated agarose beads. Then, the immunoprecipitates and cell lysates were detected with anti-
HA or anti-Flag Ab, respectively. (C) Mylipb induced the degradation of irf7 in a dose-depen-
dent manner. HEK293T cells were transfected with Myc-empty, Myc-*mylipb*, and Flag-*irf7* for
24 h, and then the cells were harvested to perform immunoblotting. (D) Wild-type (WT),
ATG5 and BECN1 knockout (KO) 293T cells were co-transfected with Myc-*mylipb* and Flag-
*irf7* for 24 h. The cell lysates were subjected to western blotting with the indicated antibodies.
(E) Mylipb induced the degradation of irf7 dependent of its ubiquitin ligase activity. HEK293T

cells were transfected with Myc-*mylipb*-WT or Myc-*mylipb*-C381S, together with HA-*irf7* for 24 h. The lysates were then subjected to IB with the indicated Abs. (F-G) Overexpression of mylipb-WT, but not mylipb-C381S suppressed the activity of ISRE reporter (F), EPC IFN reporter (G), induced by irf7.
(TIF)

**S1 Table. The primer sequences.**
(DOCX)

**S1 Data. Excel spreadsheet containing, in separate sheets, the underlying numerical data for Figs.**
(XLSX)

**S2 Data. File containing the original uncropped pictures for Figs.**
(PDF)

**S3 Data. File containing the original uncropped photos for Fig 2.**
(DOCX)

## Acknowledgments

We are grateful to Dr. Jun Cui (School of Life Sciences, Sun Yat-sen University) for providing Knockout cell lines (ATG5KO and BECN1 KO) and Guangxin Wang (Analysis and Testing Center, IHB, CAS) for her help with fluorescent microscope analysis.

## Author Contributions

**Conceptualization:** Zhi Li, Jun Li, Wuhan Xiao, Jing Wang.

**Data curation:** Zhi Li, Jun Li.

**Formal analysis:** Jun Li, Yanyi Wang.

**Funding acquisition:** Wuhan Xiao, Jing Wang.

**Investigation:** Zhi Li, Ziyi Li, Yanan Song.

**Methodology:** Chunling Wang, Le Yuan.

**Project administration:** Wuhan Xiao, Jing Wang.

**Software:** Zhi Li.

**Validation:** Zhi Li, Ziyi Li, Yanan Song.

**Visualization:** Zhi Li, Jun Li.

**Writing – original draft:** Zhi Li, Jing Wang.

**Writing – review & editing:** Wuhan Xiao, Jing Wang.

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
