## [Decision Letter · Decision Letter 0]

23 Oct 2023

Dear Dr/Associate professor wang,

Thank you very much for submitting your manuscript "Zebrafish mylipb attenuates antiviral innate immunity through two synergistic mechanisms targeting transcription factor irf3" for consideration at PLOS Pathogens. As with all papers reviewed by the journal, your manuscript was reviewed by members of the editorial board and by several independent reviewers. In light of the reviews (below this email), we would like to invite the resubmission of a significantly-revised version that takes into account the reviewers' comments.

We cannot make any decision about publication until we have seen the revised manuscript and your response to the reviewers' comments. Your revised manuscript is also likely to be sent to reviewers for further evaluation.

Sincerely,

Jacob S. Yount

Academic Editor

PLOS Pathogens

Sonja Best

Section Editor

PLOS Pathogens

Kasturi Haldar

Editor-in-Chief

PLOS Pathogens

orcid.org/0000-0001-5065-158X

Michael Malim

Editor-in-Chief

PLOS Pathogens

orcid.org/0000-0002-7699-2064

Reviewer's Responses to Questions

**Part I - Summary**

Reviewer #1: These authors report zebrafish mylipb negatively regulates IRF3-mediated antiviral innate immunity through two synergistic mechanisms. 1) Mylipb promotes K6-linked polyubiquitination and autophagic degradation of IRF3. 2) Mylipb also acts as a decoy substrate for the phosphokinase tbk1 to attenuate irf3. Although the in vivo and in vitro experiments with overexpression and KO system convincingly support the view that zebrafish mylipb negatively regulates IFN-I signaling. However, the results that refer to the proof of regulatory mechanism are superficial (Fig 4, 5, 7), and some results are inconsistent. Furthermore the western blot experiments are conducted only with overexpression system.

Reviewer #2: In this manuscript, the authors report a role of mylipb in inhibiting irf3 functions in zebrafish. Using overexpression and under-expression strategies, the results suggest that mylipb inhibits virus-induced type-I IFN production via two synergistic mechanisms - induction of autophagic degradation of irf3 and inhibition of irf3 phosphorylation. mylipb interacts with irf3 and targets Lys352 for K6-linked polyubiquitination leading to autophagy-mediated degradation. Furthermore, mylipb gets phosphorylated by tbk1 and functions as a competitive inhibitor of irf3, which also works as a substrate for tbk1. Using this mechanism mylipb acts as a decoy substrate for tbk1, thereby attenuating phosphorylation of irf3 and cellular antiviral responses. Finally, they have demonstrated the physiological role of mylipb in vivo; knockout zebrafish are more resistant to SCVC infection due to increased viral replication and antiviral gene expression levels. Overall, there are several strengths in this study, including complementary mechanistic and physiological approaches and strong biochemical analyses. However, there are some weaknesses that need to be addressed.

**Part II – Major Issues: Key Experiments Required for Acceptance**

Reviewer #1: 1. It is necessary to study the phosphorylation, stability, ubiquitination of IRF3 in mylipb-KO cells, together with the proof of the endogenous IRF3.

2. From Fig 4G to 4K. The author proved that Mylipb induces autophagic degradation of IRF3 with WB and IF experiments. Several questions arise from this:

1) In fig 4K, the IF result indicated that overexpression of mylipb induce the LC3 puncta formation. LC3 puncta formation is a sign of autophagosome flux, indicating more LC3-I have been cleaved into the LC3-II. However, in fig 4J, the amount of lc3-II did not distinctly increase when overexpressing mylipb, indicating that mylipb did not induce the autophagosome flux. So the IF and IB results are inconsistent.

2) It is skeptical that the green signal of LC3 is undetected in the control group in fig 4K. In fig 4H, the bands of LC3-I and LC3-II between control group and mylipb-transfected group are without any difference. Whether the result is representative.

Reviewer #2: 1. The specificity of mylipb for irf3 is not solid. They should test if mylipb also targets irf7 and nf-kb proteins.

2. Inflammatory genes also contribute to the viral pathogenesis and induction of some of these genes should be tested in the whole fish study.

3. The species with slower mobilities in Fig 7 for irf3 and tbk1 were considered as phosphoproteins. However, some more specific studies, e.g., either phospho-specific antibodies or phosphatase should be used to come to a reasonable conclusion.

4. In Fig 5E, wt mylipb did not degrade irf3 protein, as expected. There was no explanation for this and also the mutant mylipb was used in this expt and for functional studies in Fig 6. mylipb mutant should be tested in the ubiquitination studies as well.

5. Autophagy-mediated degradation of irf3 has only been tested using inhibitors, which have off-target effects; genetic approaches (e.g., knockdown of autophagy components) should be used for validation.

**Part III – Minor Issues: Editorial and Data Presentation Modifications**

Reviewer #1: 1. In fig 4A, the ISRE sequence is cloned from which species sample, because you have already labelled the IFN-luc with EPC or Dr. What are its sources?

2. In fig 5D, we can see that only mylipb-wt and 1-416 can induce the degradation of IRF3, while in fig 5B, the result is undetected in TCL.

3. For the ubiquitination assay result, three questions arise from this:

1) In fig 5C, IRF3 ubiquitination in △380-416 and C381S groups are still increased compared with the control group. So the result that E3 ubiquitin ligase activity was required for mylipb-mediated irf3 ubiquitination is skeptical. The ubiquitination assay of mylipb (1-280), (1-380), (1-416) should also conducted.

2) In fig 5F, the ubiquitination band is undetected in the K6R group, indicating that only K6-linked polyubiquitination chains can modify IRF3. However, in fig 5G, irf3 can be modified by any types of polyubiquitination chains. These results are contradictory.

3)It is suggested that 3-MA should appeared in ubiquitination assay results of fig 5.

4. In fig 7A, 7H, when transfected with TBK1, The HA-IRF3 has three bands, I want to know which band is unphosphorylated IRF3. If the three bands represent the whole protein level of IRF3, why both mylipb-wt and mylipb-C381S can induce the reduction of IRF3 protein level, which is contradictory with previous results.

5. In fig 7C, why the p-mylipb bands disappeared in TCL and IP sample.

6.The author declared that TBK1 phosphorylates mylipb. However, In fig 7D and E, the p-mylipb did not increase gradually with the overexpression of TBK1.

Reviewer #2: 1. In Fig 1, the levels of mylipb should be provided for the overexpression and knockout studies.

2. In Fig 2A, dose is not mentioned; also, the rationale for choosing 18h is not mentioned.

3. In Fig 3H-J the MOI of SCVC in EPC cell is missing.

4. In Figure 1B the titer of SCVC is mentioned but the MOI is missing.

5. In Figure 3D, line no 852, “After 24h, we infected (instead of transfected) the cells with SVCV”. The MOI is also not mentioned.

6. In Figure 6B and Figure 6D, MOI is missing.

PLOS authors have the option to publish the peer review history of their article (what does this mean?). If published, this will include your full peer review and any attached files.

Reviewer #1: No

Reviewer #2: No

Figure Files:

Data Requirements:

Please note that, as a condition of publication, PLOS' data policy requires that you make available all data used to draw the conclusions outlined in your manuscript. Data must be deposited in an appropriate repository, included within the body of the manuscript, or uploaded as supporting information. This includes all numerical values that were used to generate graphs, histograms etc.. For an example see here on PLOS Biology: http://www.plosbiology.org/article/info:doi%2F10.1371%2Fjournal.pbio.1001908#s5.
---

## [Editor Report · Decision Letter 1]

26 Apr 2024

Dear Dr/Associate professor wang,

We are pleased to inform you that your manuscript 'Zebrafish mylipb attenuates antiviral innate immunity through two synergistic mechanisms targeting transcription factor irf3' has been provisionally accepted for publication in PLOS Pathogens.

Best regards,

Jacob S. Yount

Academic Editor

PLOS Pathogens

Sonja Best

Section Editor

PLOS Pathogens

Michael Malim

Editor-in-Chief

PLOS Pathogens

orcid.org/0000-0002-7699-2064
---

## [Editor Report · Acceptance letter]

8 May 2024

Dear Dr/Associate professor wang,

We are delighted to inform you that your manuscript, "Zebrafish mylipb attenuates antiviral innate immunity through two synergistic mechanisms targeting transcription factor irf3," has been formally accepted for publication in PLOS Pathogens.

Best regards,

Michael Malim

Editor-in-Chief

PLOS Pathogens

orcid.org/0000-0002-7699-2064